# Natural Occurrence of Beauvericin and Enniatins in Corn- and Wheat-Based Samples Harvested in 2017 Collected from Shandong Province, China

**DOI:** 10.3390/toxins11010009

**Published:** 2018-12-27

**Authors:** Xiaomin Han, Wenjing Xu, Jing Zhang, Jin Xu, Fengqin Li

**Affiliations:** NHC Key Laboratory of Food Safety Risk Assessment, China National Center for Food Safety Risk Assessment, Beijing 100021, China; hanxiaomin@cfsa.net.cn (X.H.); xuwenjing@cfsa.net.cn (W.X.); zhangjing@cfsa.net.cn (J.Z.); xujin@cfsa.net.cn (J.X.)

**Keywords:** BEA and ENNs, corn and corn-based samples, wheat and wheat-based samples, China, HPLC-MS/MS, China

## Abstract

Totals of 158 corn and corn-based samples and 291 wheat and wheat-based samples from Shandong province, China in 2017 were analyzed for five mycotoxins including beauvericin (BEA), enniatin A (ENA), enniatin A_1_ (ENA_1_), enniatin B (ENB), and enniatin B_1_ (ENB_1_) by high-performance liquid chromatography-tandem mass spectrometry (HPLC-MS/MS). BEA was the predominant toxin detected, followed by ENB, ENA_1_, ENA, and ENB_1_. Corn and corn-based samples were more easily contaminated by BEA with an average concentration of 65.26 µg/kg, compared with that in wheat and wheat-based samples (average = 0.41 µg/kg). Concentrations of BEA, ENA, and ENB_1_ in corn kernels, flours, and flakes were significantly different (Kruskal–Wallis Test, *p* < 0.05), as well as for BEA, ENA, ENB, and ENB_1_ in wheat kernels, flours, and noodles (Kruskal–Wallis test, *p* < 0.05). Furthermore, 59.5% (94/158) and 59.8% (174/291) corn- and wheat-based samples were co-contaminated by at least two mycotoxins, respectively. Positive correlations in concentrations were observed in corn between levels of ENA and ENB_1_, ENA and ENB, ENA_1_ and ENB_1_, as well as in wheat between BEA and ENA, BEA and ENA_1_, BEA and ENB, BEA and ENB_1_, ENA and ENA_1_, ENA and ENB, ENA and ENB_1_, ENA_1_ and ENB, ENA_1_ and ENB_1_, and ENB and ENB_1_. These results demonstrate that co-contamination of BEA and enniatins (ENNs) in corn- and wheat-based samples from Shandong, China is very common. More data on the contamination of five mycotoxins in cereal and cereal-based samples nationwide are needed.

## 1. Introduction

Beauvericin (BEA) and enniatins (ENNs) are emerging mycotoxins mainly produced by fungi of the *Fusarium* genus such as *F. oxysporum*, *F. avenaceum*, *F. equiseti*, and *F. poae*. They are structurally related to cyclic hexadepsipeptides, consisting of three alternating hydroxyisovaleryl and N-methylamino acid residues [1,2]. In total, 29 naturally occurring enniatin analogs have been identified so far, but only four enniatins including enniatin A (ENA), A_1_ (ENA_1_), B (ENB), and B_1_ (ENB_1_) are frequently detected in various foods and feeds [3].

BEA and ENNs have been shown to exhibit a variety of biological properties including antibiotic, anti-insect, antifungal, and cytotoxic activities [4,5,6,7,8]. They are also reported as ionophores, enzyme inhibitors including inhibiting the enzyme acyl-CoA and cholesterol acyltransferase, and inducers of oxidative stress [8,9,10]. In addition, a lot of studies on the subject of BEA and ENNs contamination in food are available; however, most of them are focused on cereals from Europe [3,4,11,12,13,14]. High frequencies and concentrations of BEA and ENNs in cereals and cereal samples have been reported in recent years. According to Sørensen et al., 100% and 98% of maize harvested in 2006 from Denmark were contaminated with BEA and ENNs, respectively. Levels of mg/kg of BEA and ENA_1_ such as 0.51 mg/kg to 11.78 mg/kg for BEA and 33.38 mg/kg to 814.42 mg/kg for ENA_1_ were found in cereals from Spain [14]. Similarly, high concentrations of those mycotoxins were also found in breakfast cereals from Morocco [11]. Therefore, due to their diverse toxic activities and natural occurrences in food, BEA and ENNs have been receiving increasing attention. However, there has been no report concerning the natural occurrence of BEA and ENNs in China until now. The aim of this study is to evaluate the natural occurrence and distribution of BEA and ENNs in corn- and wheat-based samples from Shandong province, China.

## 2. Results

### 2.1. Natural Occurrence of BEA and ENNs in Corn- and Wheat-Based Samples

Table 1 summarized the concentrations of five mycotoxins in 158 corn and corn-based samples and 291 wheat and wheat-based samples. For 158 corn and corn-based samples, BEA was the predominant toxin with regard to the frequency and concentration. It was found that 82.3% (130/158) of samples were contaminated by BEA with the levels ranging from 0.04 μg/kg to 1006.56 μg/kg (average = 65.26 μg/kg, median = 3.88 μg/kg), followed by ENA with a positive rate of 55.1% (87/158) with the levels ranging from 0.06 μg/kg to 16.61 μg/kg (average = 0.28 μg/kg, median = 0.06 μg/kg), and ENB_1_ with a positive rate of 56.3% (89/158) and the levels ranging from 0.07 μg/kg to 3.33 μg/kg (average = 0.13 μg/kg, median = 0.14 μg/kg). Of all the 158 samples analyzed, one was contaminated with BEA at levels higher than 1000 μg/kg, two between 500 μg/kg and 1000 μg/kg, 20 between 100 μg/kg and 500 μg/kg, and 135 below 100 μg/kg. ENA_1_ and ENB were present at much lower frequencies than those detected for BEA, ENA, and ENB_1_ in all 158 analyzed samples, with a positive rate of 8.2% (13/158) and the levels ranging from 0.02 μg/kg to 6.29 μg/kg (average = 0.62 μg/kg, median = 0.15 μg/kg) for ENA_1_, and a positive rate of 3.8% (6/158) and the levels ranging from 0.05 μg/kg to 3.21 μg/kg (average = 1.19 μg/kg, median = 0.62 μg/kg) for ENB, respectively. In addition, the concentrations of BEA, ENA, and ENB_1_ were significantly different in corn kernels, flours, and flakes (Kruskal–Wallis test, *p* < 0.05); however, no significant difference was found for ENA_1_ (Kruskal–Wallis test, *p* = 0.612) or ENB (Kruskal–Wallis test, *p* = 0.850) in corn kernels, flours, and flakes. In terms of 71 corn kernels, the positive rates of five mycotoxins from highest to lowest were BEA > ENA_1_ and ENB_1_ > ENA > ENB. BEA was detected in 85.9% (61/71) samples with the levels ranging from 0.06 μg/kg to 1006.56 μg/kg (average = 46.96 μg/kg, median = 3.22 μg/kg), which was higher than those found for ENB_1_, ENA_1_, ENA, and ENB in corn kernels. In detail, the positive rate of ENB_1_ in 71 corn kernels was 4.2% (3/71) with the levels ranging from 0.10 μg/kg to 0.32 μg/kg (average = 0.21 μg/kg, median = 0.21 μg/kg), the positive rate of ENA_1_ was 4.2% (3/71) with the levels ranging from 0.09 μg/kg to 0.16 μg/kg (average = 0.14 μg/kg, median = 0.14 μg/kg), and the positive rate of ENA was 2.8% (2/71) with the levels ranging from 0.09 μg/kg to 0.17 μg/kg (average = 0.13 μg/kg, median = 0.13 μg/kg) in corn kernels. No ENB was found in 71 corn kernels. For 68 corn flour samples, the frequencies and concentrations of five mycotoxins in 68 corn flours were higher than those found in 71 corn kernels. The prevalence of five mycotoxins in 68 corn flour samples studied from highest to lowest was ENB_1_ > ENA > BEA > ENA_1_ > ENB, but the average concentration of BEA (86.45 µg/kg) was higher than those of the other four mycotoxins ENA, ENA_1_, ENB, and ENB_1_. ENB_1_ was present with the highest frequency of 98.5% (67/68) with the levels ranging from 0.08 μg/kg to 3.33 μg/kg, followed by ENA with a positive rate of 97.1% (66/68) and the levels ranging from 0.06 μg/kg to 16.61 μg/kg, and BEA with a positive rate of 95.6% (65/68) and the levels ranging from 0.04 μg/kg to 860.16 μg/kg. Furthermore, the positive rates of ENA_1_ and ENB in 68 corn flour samples were quite low with a positive rate of 14.7% (10/68) for ENA_1_ and 4.4% (3/68) for ENB. For 19 corn flakes, no ENA_1_ and ENB were detected and only four (21.1%) samples were positive for BEA; however, all 19 samples were positive for ENA and ENB_1_. It was found that the levels of BEA, ENA, and ENB_1_ in the 19 corn flakes studied were all very low with a maximum of 0.2 μg/kg for BEA, 0.07 μg/kg for ENA, and 0.13 μg/kg for ENB_1_, respectively.

For 291 wheat and wheat-based samples including 75 wheat kernels, 67 wheat flours, and 149 noodles, the positive rate of five mycotoxins from highest to lowest was ENB > BEA > ENB_1_ > ENA > ENA_1_. The detailed positive rate from highest to lowest for 291 wheat and wheat-based samples was 58.1% for ENB with the levels ranging from 0.02 μg/kg to 56.31 μg/kg (average = 1.19 μg/kg, median = 0.21 μg/kg), 56.4% for BEA with the levels ranging from 0.08 μg/kg to 11.1 μg/kg (average = 0.41 μg/kg, median = 0.22 μg/kg), 54.6% for ENB_1_ with the levels ranging from 0.10 μg/kg to 61.77 μg/kg (average = 1.37 μg/kg, median = 0.32 μg/kg), 37.5% for ENA with the levels ranging from 0.09 μg/kg to 3.30 μg/kg (average = 0.19 μg/kg, median = 0.12 μg/kg), and 22.0% for ENA_1_ with the levels ranging from 0.10 μg/kg to 20.91 μg/kg (average = 0.96 μg/kg, median = 0.22 μg/kg). It was also found that the contamination of BEA, ENA, ENB, and ENB_1_ in wheat and wheat-based samples varied by type (Kruskal–Wallis test, *p* < 0.05), but not for ENA_1_ (Kruskal–Wallis test, *p* = 0.122)**.** For the 75 wheat kernels studied, the positive rate of five mycotoxins from highest to lowest was BEA (48.0%) > ENB (16.0%) > ENB_1_ (13.3%) > ENA (12.0%) and ENA_1_ (12.0%). It should be noted that there were three samples with ENB, three samples with ENB_1_, and one sample with ENA_1_ at levels higher than 20 μg/kg, with the maximum of 56.31 μg/kg (average = 12.55 μg/kg, median = 2.17 μg/kg) for ENB, 61.77 μg/kg (average = 15.61 μg/kg, median = 3.75 μg/kg) for ENB_1_, and 20.91 μg/kg (average = 5.33 μg/kg, median = 1.17 μg/kg) for ENA_1_, respectively. Of the 67 wheat flour samples analyzed, no ENA_1_ was found in 67 samples, and the maximum levels of the five mycotoxins were all lower than 20 μg/kg. BEA was found in 52 out of 67 samples at levels ranging from 0.12 μg/kg to 11.1 μg/kg (average = 0.46 μg/kg, median = 0.21 μg/kg), followed by ENB with the positive rate of 56.7% and the levels ranging from 0.02 μg/kg to 0.59 μg/kg (average = 0.18 μg/kg, median = 0.13 μg/kg), ENA with the positive rate of 52.2% and the levels ranging from 0.09 μg/kg to 0.19 μg/kg (average = 0.11 μg/kg, median = 0.10 μg/kg), and ENB_1_ with the positive rate of 49.3% and the levels ranging from 0.14 μg/kg to 1.31 μg/kg (average = 0.34 μg/kg, median = 0.27 μg/kg). Whereas, the concentrations of the five mycotoxins in 149 noodle samples were all lower than 20 μg/kg with the highest frequency of 79.9% for ENB.

### 2.2. Co-Contamination of BEA and ENNs in Corn- and Wheat-Based Samples

The co-contamination of five mycotoxins in 158 corn and corn-based samples collected from Shandong province is given in Table 2. It was found that only 10 samples (6.33%, 10/158) were not contaminated with any of the five mycotoxins studied and 54 samples (34.18%, 54/158) were only contaminated by one of the mycotoxins studied, respectively. There were 24 samples (15.19%, 24/158), 60 samples (37.97%, 60/158), seven samples (4.43%, 7/158), and three samples (1.90%, 3/158) contaminated with two mycotoxins, three mycotoxins, four mycotoxins, and five mycotoxins, respectively. The natural occurrences of the five mycotoxins differed among different corn samples, which was shown in the following details. For 71 corn kernels, BEA was the most predominant mycotoxin with a frequency of 76.06% (54/71), followed by the combinations of BEA-ENB_1_ with a frequency of 4.23% (3/71), BEA-ENA_1_-ENB with a frequency of 2.81% (2/71), and both BEA-ENA-ENA_1_ and BEA-ENA-ENB with a frequency of 1.41% (1/71), respectively. In terms of the 68 corn flours, BEA-ENA-ENA_1_ was the most predominant combination with a frequency of 76.47% (52/68), followed by the combination of four mycotoxins BEA-ENA-ENA_1_-ENB_1_ (10.29%, 7/68), two mycotoxins combination (8.82%, 6/68) including BEA-ENA_1_ (2.94%, 2/68), BEA-ENA (1.47%, 1/68) and ENA-ENA_1_ (4.41%, 3/68), and five mycotoxins combination (4.41%, 3/68). For 19 corn flakes, there was no co-occurrence of four mycotoxins and five mycotoxins. ENA-ENA_1_ and BEA-ENA-ENA_1_ combinations were found in 15 (78.95%) and four (21.05%) samples, respectively. Moreover, in corn and corn-based samples, significant correlations were observed between concentrations of ENA and ENB_1_ (*r* = 0.863, *p* < 0.01), ENA and ENB (*r* = 0.998, *p* < 0.01), and ENA_1_ and ENB_1_ (*r* = 0.887, *p* < 0.01).

The co-occurrence of five cyclic depsipeptide mycotoxins in 291 wheat and wheat-based samples is listed in Table 3. In total, 15.46% (45/291), 15.12% (44/291), 16.84% (49/291), and 13.06% (38/291) of the samples analyzed were co-contaminated by two mycotoxins, three mycotoxins, four mycotoxins, and five mycotoxins, respectively. For 75 wheat kernel samples, nearly half (46.67%, 35/75) the samples were negative for five mycotoxins, and 37.33% (28/75) samples were contaminated by only one mycotoxin (26 for BEA, one for ENA, and one for ENB alone), respectively. BEA-ENA-ENA_1_-ENB-ENB_1_ was the most frequent mycotoxin combination with a frequency of 8% (6/75), followed by BEA-ENA_1_-ENB-ENB_1_ (1.33%, 1/75), ENA-ENA_1_-ENB-ENB_1_ (1.33%, 1/75), BEA-ENB-ENB_1_ (1.33%, 1/75), ENA_1_-ENB-ENB_1_ (1.33%, 1/75), BEA-ENA (1.33%, 1/75) and BEA-ENB (1.33%, 1/75) in 75 wheat kernels. In terms of 67 wheat flour samples, there were 10 mycotoxin combinations, as shown in Table 3. BEA-ENA-ENB-ENB_1_ was the most frequent mycotoxin combination type with a frequency of 25.37% (17/67), followed by BEA-ENB-ENB_1_ with a frequency of 14.93% (10/67), and BEA-ENA with a frequency of 11.94% (8/133). For 149 noodles samples analyzed, there were 21.48% (32/149), 20.13% (30/149), 18.79% (28/149), and 18.12% (27/149) samples co-contaminated with five, four, two, and three mycotoxins, respectively. BEA-ENA-ENA_1_-ENB-ENB_1_ was the most frequent combination with a frequency of 21.48% (32/149) followed by ENB-ENB_1_ (17.45%, 26/149), BEA-ENB-ENB_1_ (10.07%, 15/149), and BEA-ENA-ENB-ENB_1_ (8.05%, 12/149). The significant linear correlations in the concentrations were observed between BEA and ENA (*r* = 0.425, *p* < 0.01), BEA and ENA_1_ (*r* = 0.592, *p* < 0.01), BEA and ENB (*r* = 0.278, *p* < 0.01), BEA and ENB_1_ (*r* = 0.237, *p* < 0.05), ENA and ENA_1_ (*r* = 0.910, *p* < 0.01), ENA and ENB (*r* = 0.563, *p* < 0.01), ENA and ENB_1_ (*r* = 0.640, *p* < 0.01), ENA_1_ and ENB (*r* = 0.827, *p* < 0.01), ENA_1_ and ENB_1_ (*r* = 0.856, *p* < 0.01), and ENB and ENB_1_ (*r* = 0.916, *p* < 0.01) in wheat and wheat-based samples, respectively.

## 3. Discussion

To the best of our knowledge, this is the first report on the natural co-occurrence of BEA, ENA, ENA_1_, ENB, and ENB_1_ in corn- and wheat-based samples from China. The maximum concentrations of BEA in corn kernel, flour, and flake samples in our study were far higher than those of four ENNs, which were different from those reported by Sørensen et al. for 73 corn samples collected from Demark in 2005 and 2006; ENB (2598 μg/kg) was much higher than BEA (73 μg/kg) [13]. In addition, the positive rates of BEA in corn kernels (85.9%), flours (95.6%), and flakes (21.1%) in our study were higher than those reported for 14 corn and 22 corn-based samples (14% and 4.5%, respectively) collected from the Mediterranean [15]. However, the frequencies of three ENNs (8.2% for ENA_1_, 3.8% for ENB, and 56.3% for ENB_1_) in all corn and corn-based samples in our study were lower than those reported by Uhlig et al. in 2000 and 2002 [12], which were 67% for ENA_1_, 100% for ENB, 94% for ENB_1_ in Norwegian wheat and wheat-based samples, respectively. Furthermore, a similar trend was also found in the wheat and wheat-based samples studied, and the levels of five mycotoxins in wheat kernels, flours, and noodles were quite low with the maximum levels ranging from 11.1 μg/kg to 61.77 μg/kg. These levels were lower than those found in Norwegian grain samples from 2001 and 2002 ranging from 58 μg/kg to 7400 μg/kg [12]. The reason for this might be due to the difference of the community ecology of fungal pathogens between Norway and China. Several reports have indicated that in the cooler maritime of Northwestern Europe, commonly occurring *Fusarium* head blight (FHB) species were *F. culmorum*, *F. avenaceum*, *Microdochium* species, while *F. asiaticum* had its main distribution in Asia (China, Japan, Nepal, and Korea) [16,17,18]. Importantly, the main strains found to produce BEA and ENNs so far are *F. culmorum* and *F. avenaceum*, and not *F. asiaticum*. 

We found that concentrations of BEA in corn and corn-based samples were far higher than those in wheat and wheat-based samples. The reason might be that corn- and wheat-based samples can be infected by different *Fusarium* species with different cyclic hexadepsipeptides toxin-producing abilities [19,20,21,22,23,24,25]. Besides, co-contamination by more than one of the five mycotoxins was common in the corn- and wheat-based samples analyzed in the present study, occurring in 59.5% (94/158) and 59.8% (174/291) of the corn- and wheat-based samples, respectively. These results were consistent with reports on BEA and ENNs contaminations of wheat samples collected from the Czech Republic and Italy [26,27].

As far as we know, this is the first report concerning BEA and ENNs co-contamination in Chinese cereal and cereal-based samples. The European Food Safety Authority (EFSA) has given a preliminary scientific opinion on the risks to animal and public health related to dietary exposure to five cyclic depsipeptide mycotoxins due to lack of sufficient data regarding the toxigenic ability of *Fusarium* species and contaminations of the five mycotoxins in cereal and cereal-based samples. However, the limitation of the present study was that the samples analyzed were only collected from Shandong province, China. To provide accurate data regarding mycotoxin contamination and the toxin-producing ability of *Fusarium* species and their biosynthetic genetic differences, there is a need for more samples to be collected nationwide across China.

## 4. Materials and Methods

### 4.1. Chemicals and Reagents

Standard powders of BEA, ENA, ENA_1_, ENB, and ENB_1_ with purities ≥97% were obtained from BioAustralis (Smithfield, NSW, Australia). Acetonitrile (ACN) and methanol (MeOH), both of LC-MS grade, were obtained from Fisher Scientific (Fair Lawn, NJ, USA). Ammonium acetate purchased from Sigma-Aldrich (St. Louis, MO, USA) was of MS grade. Water was purified successively by a Millipore Milli-Q system (Millipore, Bedford, MA, USA) with a conductivity ≥18.2 MΩ.cm at 25 °C.

### 4.2. Sample Collection

A total of 158 corn-based samples including 71 kernels, 68 flours, and 19 flakes, and 291 wheat-based samples including 75 kernels, 67 flours, and 149 noodles were randomly collected in different districts of Shandong provinces, an area which was the most important grain planting and consuming province in China. All these corn- and wheat-based samples were intended for human consumption and collected from a supermarket, farmer’s market, or local resident’s home. Three subsamples were taken from the top, middle, and bottom layers of the container (five sampling sites for each layer). For wheat flour and corn flour samples, 5 g of each were weighed for toxin extraction directly after mixing. For corn kernel, corn flake, wheat kernel, and noodles, the subsamples were pooled and mixed thoroughly using a grinder (VICAM, Nixa, MO, USA), and approximately 200 g of the mixture was finely ground to a powder and kept in Ziploc plastic bags, sealed, and stored at −20 °C prior to analysis. A 5 g finely ground portion of each sample was weighed for analysis.

### 4.3. Toxin Analysis

All five mycotoxins were extracted according to the method reported previously [28]. A total of 5 g of each were extracted with 40 mL of acetonitrile-water (85:15, *v*/*v*) by blending for 30s to 60s using a vortex and then shaking for 30 min. The extract was transferred to a new 50 mL centrifuge tube and allowed to stand for 15–20 min under room temperature. Each aliquot of 10 mL of supernatant was diluted with 20 mL of distilled water followed by homogenization using a vortex. A quantity of 4 mL of the diluted extract was applied to a Sep-Pak Vac 3cc (200 mg) C_18_ Cartridge (Waters, Milford, MA, USA), which was pre-equilibrated with 3 mL of methanol and 3 mL of water. The cartridge was washed with 3 mL of 10% acetonitrile in water followed by a wash with 3 mL of 50% acetonitrile in water. Toxins were eluted using 2 mL of 90% acetonitrile in water.

### 4.4. UPLC Conditions

UPLC/ESI-MS/MS system equipped with ExionLC (SHIMADZU, Kyoto, Japan), QTRAP^TM^ 5500 MS/MS system (AB Sciex, Foster City, CA, USA) and a MultiQuant^TM^ Version 3.0.2 software (AB Sciex, Foster City, CA, USA) for data acquisition and analysis was used to quantify 5 mycotoxins in positive mode. Chromatographic separation was achieved using a C_18_ column (2.1 mm × 50 mm, 1.7 μm bead diameter, Waters, Milford, MA, USA). Temperatures of the UPLC column and autosampler were set at 35 °C and 15 °C, respectively. The mobile phase A was water containing 2 mmol/L ammonium acetate, and the mobile phase B was acetonitrile. The program was starting from B/A (0/100, *v*/*v*) in the first 2 min, reached B/A (60/40, *v*/*v*) in 2 min to 3 min, and continued to decrease B/A (70/30, *v*/*v*) in 3 min to 19 min. Afterward, A was linearly increased to 100% (B/A, 0/100, *v*/*v*) within 2 min and maintained at this composition of the mobile phase for 0.1 min.

### 4.5. MS/MS Conditions

Data acquisition was performed by applying multiple reaction monitoring (MRM) with an optimized dwell time of 200 ms in order to obtain the optimal sensitivity and selectivity of MS conditions in positive electrospray ionization (ESI^+^) mode. The curtain gas was set to 30 psi, the collision gas to medium, Gas 1 to 80 psi, and Gas 2 to 80 psi. The source desolvation temperature was set at 550°C. The ion spray voltage was set to +5500V. Parent and fragment ions (quantifier and qualifier) for each analyte were chosen based on the best signal-to-noise ratios in a blank spiked sample. Detailed information for the parent and fragment ions is shown in Appendix A.

### 4.6. Method Validation

To evaluate the method proficiency, BEA- and ENNs-free corn- and wheat-based samples were spiked with five mycotoxins at three concentrations. The spiked samples (*n* = 6 repeats each level) were extracted and analyzed for recovery calculation. Mean recoveries, in which the matrix effect was compensated, were in the ranges of 91.6–129.0% (BEA), 95.8–140.5% (ENA), 97.1–139.3% (ENA_1_), 106.5–149.7% (ENB), and 102.7–142.2% (ENB_1_) as determined from six parallel analyses of blank samples spiked with 50 to 250 µg/kg for BEA, from 7.5 to 37.5 µg/kg for ENA, from 20 to 100 µg/kg for ENA_1_, from 20-100 µg/kg for ENB, and from 50 to 250 µg/kg for ENB_1_ with the relative standard deviations (RSD) of 1.5–7.7% for BEA, 0.4–12.9% for ENA, 1.4–6.9% for ENA_1_, 1.7–13.8% for ENB, and 1.2–12.6% for ENB_1_, respectively. The limits of detection and limits of quantification for BEA, ENA, ENA_1_, ENB, and ENB_1_ in different types of samples are presented in Table 4. The spiked samples were extracted and analyzed five times within one day for intra-day precision or analyzed in five successive days according to the extracted method described above for inter-day precision. The data obtained were used for RSD calculation and all the RSD for the intra- and inter-day precision were less than 15%.

### 4.7. Data Analysis

All parameters including positive rate, average, median, and range of BEA and ENNs in the analyzed samples were obtained by applying the SPSS statistical package (version 20.0, IBM, USA). The Kruskal–Wallis test was employed for statistical analysis by comparing the mycotoxin concentrations in different samples.

## Figures and Tables

**Table 1 toxins-11-00009-t001:** Natural occurrence of beauvericin (BEA) and enniatins (ENNs) in corn- and wheat-based samples from Shandong province.

Sample Type	Sample	Mycotoxin	Occurrence % (*n*)	Range (μg/kg)	Average (μg/kg)	Median (μg/kg)
**Corn and Corn-Based Samples**	Corn kernel (71)	BEA	85.9 (61)	0.06–1006.56	46.96	3.22
		ENA ^a^	2.8 (2)	0.09–0.17	0.13	0.13
		ENA_1_ ^b^	4.2 (3)	0.09–0.16	0.14	0.14
		ENB ^c^	0			
		ENB_1_ ^d^	4.2 (3)	0.10–0.32	0.21	0.21
	Corn flour (68)	BEA	95.6 (65)	0.04–860.16	86.45	7.32
		ENA	97.1 (66)	0.06–16.61	0.35	0.06
		ENA_1_	14.7 (10)	0.02–6.29	0.76	0.15
		ENB	4.4 (3)	0.86–3.21	2.15	2.37
		ENB_1_	98.5 (67)	0.08–3.33	0.15	0.1
	Corn flake (19)	BEA	21.1 (4)	0.1–0.2	0.17	0.17
		ENA	100 (19)	0.06–0.07	0.06	0.06
		ENA_1_	0			
		ENB	0			
		ENB_1_	100 (19)	0.07–0.13	0.08	0.09
	Total (158)	BEA	82.3 (130)	0.04–1006.56	65.26	3.88
		ENA	55.1 (87)	0.06–16.61	0.28	0.06
		ENA_1_	8.2 (13)	0.02–6.29	0.62	0.15
		ENB	3.8 (6)	0.05–3.21	1.19	0.62
		ENB_1_	56.3 (89)	0.07–3.33	0.13	0.14
**Wheat and Wheat-Based Samples**	Wheat kernel (75)	BEA	48.0 (36)	0.08–3.45	0.68	0.45
		ENA	12.0 (9)	0.19–3.30	0.87	0.29
		ENA_1_	12.0 (9)	0.25–20.91	5.33	1.17
		ENB	16.0 (12)	0.38–56.31	12.55	2.17
		ENB_1_	13.3 (10)	0.50–61.77	15.61	3.75
	Wheat flour (67)	BEA	77.6 (52)	0.12–11.1	0.46	0.21
		ENA	52.2 (35)	0.09–0.19	0.11	0.1
		ENA_1_	0			
		ENB	56.7 (38)	0.02–0.59	0.18	0.13
		ENB_1_	49.3 (33)	0.14–1.31	0.34	0.27
	Noodles (149)	BEA	51.0 (76)	0.12–1.22	0.24	0.2
		ENA	43.6 (65)	0.09–0.75	0.14	0.12
		ENA_1_	36.9 (55)	0.10–1.19	0.25	0.16
		ENB	79.9 (119)	0.02–2.41	0.37	0.22
		ENB_1_	77.9 (116)	0.10–2.64	0.44	0.31
	Total (291)	BEA	56.4 (164)	0.08–11.1	0.41	0.22
		ENA	37.5 (109)	0.09–3.30	0.19	0.12
		ENA_1_	22.0 (64)	0.10–20.91	0.96	0.22
		ENB	58.1 (169)	0.02–56.31	1.19	0.21
		ENB_1_	54.6 (159)	0.10–61.77	1.37	0.32

^a^: ENA = enniatin A; ^b^: ENA_1_ = enniatin A_1_; ^b^: ENB = enniatin B; ^d^: ENB_1_ = enniatin B_1._

**Table 2 toxins-11-00009-t002:** Co-occurrence of BEA and ENNs in corn and corn-based samples collected from Shandong (*n* = 158).

Contamination by	Corn Kernel (*n* = 71)		Corn Flour (*n* = 68)		Corn Flake (*n* = 19)		Occurrence Sum % (*n*)
	Occurrence % (*n*)	Toxin Combination	Occurrence % (*n*)	Toxin Combination	Occurrence % (*n*)	Toxin Combination	
Two Mycotoxins	4.23 (3)	BEA-ENB_1_ (4.23%, 3/71)	8.82 (6)	BEA-ENA_1_ (2.94%, 2/68)	78.95 (15)	ENA-ENA_1_ (78.95%,15/19)	15.19 (24)
				BEA-ENA (1.47%, 1/68)			
				ENA-ENA_1_ (4.41%, 3/68)			
Three Mycotoxins	5.63 (4)	BEA-ENA-ENA_1_ (1.41%, 1/71)	76.47 (52)	BEA-ENA-ENA_1_ (76.47%, 52/68)	21.05 (4)	BEA-ENA-ENA_1_ (21.05%, 4/19)	37.97 (60)
		BEA-ENA_1_-ENB (2.81%, 2/71)					
		BEA-ENA-ENB (1.41%, 1/71)					
Four Mycotoxins	0		10.29 (7)	BEA-ENA-ENA_1_-ENB (10.29%, 7/68)	0		4.43 (7)
Five Mycotoxins	0		4.41 (3)	BEA-ENA-ENA_1_-ENB-ENB_1_ (4.41%, 3/68)	0		1.90 (3)

**Table 3 toxins-11-00009-t003:** Co-occurrence of BEA and ENNs in wheat and wheat-based samples collected from Shandong province (*n* = 291).

Contamination By	Wheat Seed (*n* = 75)		Wheat Flour (*n* = 67)		Noodles (*n* = 149)		Occurrence Sum % (*n*)
	Occurrence % (*n*)	Toxin Combination	Occurrence % (*n*)	Toxin Combination	Occurrence % (*n*)	Toxin Combination	
Two Mycotoxins	2.67 (2)	BEA-ENA (1.33%, 1/75)	22.39 (15)	BEA-ENA (11.94%, 8/67)	18.79 (28)	BEA-ENA (1.34%, 2/149)	15.46 (45)
		BEA-ENB (1.33%, 1/75)		BEA-ENB (2.99%, 2/67)		ENB-ENB_1_ (17.45%, 26/149)	
				ENA-ENB (1.49%, 1/67)			
				ENA-ENB_1_ (2.99%, 2/67)			
				ENB-ENB_1_ (2.99%, 2/67)			
Three Mycotoxins	2.67 (2)	BEA-ENB-ENB_1_ (1.33%, 1/75)	22.39 (15)	BEA-ENB-ENB_1_ (14.93%, 10/67)	18.12 (27)	BEA-ENB-ENB_1_ (10.07%, 15/149)	15.12 (44)
		ENA_1_-ENB-ENB_1_ (1.33%, 1/75)		BEA-ENA-ENB (4.48%, 3/67)		ENA-ENB-ENB_1_ (4.70%, 7/149)	
				BEA-ENA-ENB_1_ (1.49%, 1/67)		ENA_1_-ENB-ENB_1_ (3.36%, 5/149)	
				ENA-ENB-ENB_1_ (1.49%, 1/67)			
Four Mycotoxins	2.67 (2)	BEA-ENA_1_-ENB-ENB_1_ (1.33%, 1/75)	25.37 (17)	BEA-ENA-ENB-ENB_1_ (25.37%, 17/67)	20.13 (30)	BEA-ENA_1_-ENB-ENB_1_ (5.37%, 8/149)	16.84 (49)
		ENA-ENA_1_-ENB-ENB_1_ (1.33%, 1/75)				ENA-ENA_1_-ENB-ENB_1_ (6.71%, 10/149)	
						BEA-ENA-ENB-ENB_1_ (8.05%, 12/149)	
Five Mycotoxins	8.00 (6)	BEA-ENA-ENA_1_-ENB-ENB_1_ (8.00%, 6/75)	0		21.48 (32)	BEA-ENA-ENA_1_-ENB-ENB_1_ (21.48%, 32/149)	13.06 (38)

**Table 4 toxins-11-00009-t004:** The limit of detection and limit of quantity for BEA and ENNs in corn- and wheat-based samples (µg/kg).

Mycotoxin	Corn		Corn-Based Samples		Wheat		Wheat-Based Samples	
	LOD ^f^	LOQ ^g^	LOD	LOQ	LOD	LOQ	LOD	LOQ
BEA ^a^	0.04	0.12	0.03	0.09	0.08	0.26	0.02	0.08
ENA ^b^	0.02	0.07	0.01	0.02	0.06	0.21	0.03	0.10
ENA_1_ ^c^	0.09	0.42	0.12	0.44	0.21	0.68	0.10	0.33
ENB ^d^	0.01	0.03	0.02	0.06	0.02	0.05	0.02	0.07
ENB_1_ ^e^	0.10	0.33	0.08	0.28	0.14	0.48	0.10	0.32

^a^: BEA = beauvericin; ^b^: ENA = enniatin A; ^c^: ENA_1_ = enniatin A_1_; ^d^: ENB = enniatin B; ^e^: ENB_1_ = enniatin B_1_; ^f^: LOD = limit of detection; ^g^: LOQ = limit of quantity.

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
