# Peer review of "Natural Occurrence of Beauvericin and Enniatins in Corn- and Wheat-Based Samples Harvested in 2017 Collected from Shandong Province, China"

_toxins, 2018, doi:10.3390/toxins11010009_

Round 1

Reviewer 1 Report

This manuscript, written in excellent English, requires only minor revisions.

Page 3. Check the heading of Table 1. In the 3rd column it should be written Positive in the first line and in the second Rate (%). In the 4th column (Range) µg kg-1 should be added.

Page 6. Table 2. In the heading second, fourth, fifth and seventh column should be changed because in the columns first were given percentages and then the number of positive samples.

Page 8. The same changes should be done in the Table 3.

Author Response

This manuscript, written in excellent English, requires only minor revisions.1. Page 3. Check the heading of Table 1. In the 3rd column it should be written Positive in the first line and in the second Rate (%). In the 4th column (Range ) µg kg-1 should be added.Response: We agree with the suggestion from reviewer 1, and we have re-written the heading of Table 1 as suggested by reviewer 1. 2. Page 6. Table 2. In the heading second, fourth, fifth and seventh column should be changed because in the columns first were given percentages and then the number of positive samples.Response: We agree with the suggestion from reviewer 1, and we have re-written the heading of Table 2 as suggested by reviewer 1. 
3. Page 8. The same changes should be done in the Table 3.Response: We agree with the suggestion from reviewer 1, and we have re-written the heading of Table 3 as suggested by reviewer 1.

Reviewer 2 Report

The results section presents results in both text and table and is difficult to read and comprehend.  The text should be shortened to give only the major points of interest, and then should refer to the tables for the reader to obtain fine detail. The tables should have more precise headings.  "Mycotoxin" (no "s"), "No. positive (%) and he last three must have units (ug/kg) in brackets. Line 120-121 is an unintended double negative.  Should read ".. were not contaminated with any of the five mycotoxins.."  Line 210 "Bioaustralis (Smithfield, NSW, Australia)".  The paragraph lines 185-197 should be restructured.  The reason why BEA in corn is higher than in wheat is simple: different Fusarium species grow in maize and wheat nd prodcure different toxins.  That should be stated clearly: the topic has nothing to do with biochemical precursors.  Line 266.  Limits for detection and quantification should be in the main paper for everyone to see, not in Supplementary Information.

Author Response

Responses to the reviewers' commentsTO reviewer #2:

1. The text should be shortened to give only the major points of interest, and then should refer to 

the tables for the reader to obtain fine detail.

Response: It is a good suggestion to shorten the manuscript and only give the major

points of interest, and we have re-written some sentences of manuscript.It was

listed in line 84 to line 87,line 119 to 120, line 136, line 189 to line 207 in resubmitted tracking version.

2. The tables should have more precise headings.  "Mycotoxin" (no "s"), "No. positive (%) and the last three must have units (ug/kg) in brackets.

Response: As suggested, the headings of tables 1, table 2 and table 3 were re-written such as deleting the “s” in “Mycotoxins”, adding the units (ug/kg) in brackets, and the expression of occurrence and number.

3. Line 120-121 is an unintended double negative.  Should read ". were not contaminated with any of the five mycotoxins.."

Response: Yes, as suggested in the revised version, we have changed the word “none” to “any”. It was shown as follows in red font.It was found that only 10 samples (6.33%, 10/158) were not contaminated with any of 5 mycotoxins studied and 54 samples (34.18%, 54/158) were only contaminated by one mycotoxin studied, respectively.

4. Line 210 "Bioaustralis (Smithfield, NSW, Australia)".

Response: Yes, as suggested in the revised version, we have changed as follows.BioAustralis (Smithfield, NSW, Australia).

5. The paragraph lines 185-197 should be restructured.  The reason why BEA in corn is higher than in wheat is simple: different Fusarium species grow in maize and wheat nd prodcure different toxins.  That should be stated clearly: the topic has nothing to do with biochemical precursors.

Response: Yes, we agree with the description about the reason why BEA in corn is higher than in wheat, and it was listed as follows.We found that concentrations of BEA in corn and corn-based samples were far higher than those in wheat and wheat-based samples. The reason might be

that corn- and wheat based samples can be infected by different Fusarium species with different

cyclic hexadepsipeptides toxin-producing abilities [19-25].

6. Line 266.  Limits for detection and quantification should be in the main paper for everyone to see,

not in Supplementary Information.

Response: Yes, as suggested in the revised version, we have moved the Supplementary Table S2 to the main paper as Table 4.
